# AL-GTD: Deep Active Learning for Gaze Target Detection

Francesco Tonini
University of Trento
Fondazione Bruno Kessler
Trento, Italy
francesco.tonini@unitn.it

Nicola Dall'Asen
University of Trento
University of Pisa
Pisa, Italy
nicola.dallasen@unitn.it

Lorenzo Vaquero
Fondazione Bruno Kessler
Trento, Italy
lvaquerootal@fbk.eu

Cigdem Beyan
Department of Computer Science
University of Verona
Verona, Italy
cigdem.beyan@univr.it

Elisa Ricci
University of Trento
Fondazione Bruno Kessler
Trento, Italy
e.ricci@unitn.it

## ABSTRACT

Gaze target detection aims at determining the image location where a person is looking. While existing studies have made significant progress in this area by regressing accurate gaze heatmaps, these achievements have largely relied on access to extensive labeled datasets, which demands substantial human labor. In this paper, our goal is to reduce the reliance on the size of labeled training data for gaze target detection. To achieve this, we propose AL-GTD, an innovative approach that integrates supervised and self-supervised losses within a novel sample acquisition function to perform active learning (AL). Additionally, it utilizes pseudo-labeling to mitigate distribution shifts during the training phase. AL-GTD achieves the best of all AUC results by utilizing only 40-50% of the training data, in contrast to state-of-the-art (SOTA) gaze target detectors requiring the entire training dataset to achieve the same performance. Importantly, AL-GTD quickly reaches satisfactory performance with 10-20% of the training data, showing the effectiveness of our acquisition function, which is able to acquire the most informative samples. We provide a comprehensive experimental analysis by adapting several AL methods for the task. AL-GTD outperforms AL competitors, simultaneously exhibiting superior performance compared to SOTA gaze target detectors when all are trained within a low-data regime. Code is available at https://github.com/francescotonini/al-gtd.

## CCS CONCEPTS

• **Human-centered computing** → **Collaborative and social computing**; • **Computing methodologies** → **Machine learning**.

## KEYWORDS

Gaze target detection, active learning, social signals, human-human interaction, multimodal data

**ACM Reference Format:**
Francesco Tonini, Nicola Dall'Asen, Lorenzo Vaquero, Cigdem Beyan, and Elisa Ricci. 2024. AL-GTD: Deep Active Learning for Gaze Target Detection. In *Proceedings of the 32nd ACM International Conference on Multimedia (MM '24), October 28-November 1, 2024, Melbourne, VIC, AustraliaProceedings of the 32nd ACM International Conference on Multimedia (MM'24), October 28-November 1, 2024, Melbourne, Australia.* ACM, New York, NY, USA, 10 pages. https://doi.org/10.1145/3664647.3680952

## 1 INTRODUCTION

Human communication relies on a range of multimodal cues, including speech and gestures. Among these cues, the gaze holds significant importance as it reveals a person's visual focus of attention, enabling us to comprehend the interests, intentions, or (future) actions of individuals [21]. Gaze analysis has been extensively utilized across various fields, including human-computer interaction [6, 49], neuroscience [16, 52], social robotics [1], and social and organizational psychology [9, 19]. Automated gaze behavior analysis has been addressed through two tasks: *gaze estimation* and *gaze target detection*. Gaze estimation involves discerning the direction of a person's gaze, usually within a 3D space, when provided with a cropped human head image as input. This encompasses estimating the horizontal and vertical angles of the gaze, as well as determining the depth or distance at which the gaze is focused [11, 31, 36]. On the other hand, gaze target detection (or gaze-following) is the process of identifying the specific point or area within a scene that a person is looking at [15, 33, 60]. This involves analyzing visual cues such as head orientation, and other contextual information to determine the focal point of a person's gaze [4, 14, 15, 23, 34, 41, 47, 53, 59–61].

In this study, we focus on the gaze target detection task. Our multimodal gaze network, namely **GTN**, integrates the head crop of the person of interest with RGB and depth data. We employ two separate backbones for processing scene RGB and depth data to extract meaningful features indicating areas of interest. A third backbone is used for head processing to predict an attention map projected into the scene features, which identifies image areas where the person is more likely to look. Then, the fusion of head, scene and depth features generates the final gaze heatmap, centered on the person's visual focus of attention.

The training and evaluation of the gaze target detection task are carried out using manually human-annotated data. In certain

Francesco Tonini, Nicola Dall'Asen, Lorenzo Vaquero, Cigdem Beyan, and Elisa Ricci

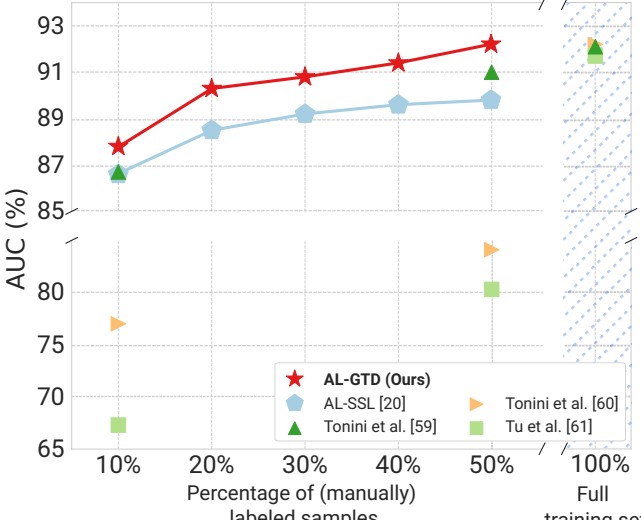

**Figure 1: The performance of our AL-GTD compared to counterpart active learning approach AL-SSL [20] and SOTA gaze target detectors: Tu et al. [61] and Tonini et al. [59, 60] on the GazeFollow dataset [54]. Our method consistently performs better than competitors and achieves SOTA AUC performance with half of the training data, proving the effectiveness of our acquisition function.**

instances, collecting a gaze target detection dataset proves to be challenging, leading to low consensus among annotators, as shown in [47, 60]. Moreover, manual labeling is a highly tedious and time-consuming process. Some studies have reported labor times ranging from ten seconds to one minute for every second of gaze data [3, 17]. Another study [22] revealed that an experienced human annotator may require two to more than ten times the duration of a video to accurately label the gaze. Unfortunately, the performance of state-of-the-art (SOTA) gaze target detectors, especially those relying on a Transformer architecture [60, 61], mainly depends on the size of the labeled training set.

In this paper, our objective is to explore solutions for reducing reliance on the size of labeled training data in gaze target detection. In essence, given our method GTN, we aim to find ways to achieve accurate performance even with a smaller amount of labeled training data. The key is to effectively select a subset of the data that optimizes model training. In this vein, researchers have delved into various strategies for choosing the most informative samples in a dataset for labeling, a methodology commonly referred to as Active Learning (AL). Our study focuses on AL's pioneering application in gaze target detection. We opt to keep our gaze detector relatively simple (*e.g.*, we do not include body pose and scene point clouds [4] that can potentially improve the results, or perform extra tasks like retrieving the gazed-object's location [60]), yet we still prove our GTN's effectiveness, particularly in a low data regime.

Usually, AL is achieved by developing a scoring function, *i.e.* the *AL acquisition function*, that selects the most informative samples, which are the ones contributing the most to the effectiveness of the model. This function, for example, can prompt the labeling of samples for which the network exhibits the highest uncertainty, indicating its lowest confidence in predictions [5, 26, 38, 58, 65].

Other popular AL acquisition functions are based on diversity [18, 56] or ensemble of multiple learning models [5]. Instead, we present a novel AL acquisition function, which is integrated into our GTN.

Our method, named **AL-GTD**, utilizes three metrics to assess the informativeness of the network predictions on unlabeled samples. These scores measure (a) the discrepancy and (b) scatteredness between the network's intermediate attention map and predicted gaze heatmap, as well as (c) the scene content described by an object detector. To further enhance the performance, we incorporate a self-supervised learning (SSL) strategy that measures the consistency of network predictions against image augmentations. Furthermore, inspired by [20], we include pseudo-labeling in our pipeline, and automatically annotate unlabeled samples using the prediction of GTN. Pseudo-labeling enables us to expand the training dataset without incurring additional labeling costs while also handling distribution shifts resulting from the acquisition function's objective.

Given that this study marks the initial endeavor of integrating gaze target detection within AL, we also incorporate several existing AL acquisition functions applied to other tasks to conduct a comparative study to evaluate the effectiveness of our AL-GTD. To this end, we leverage the gaze target detection datasets GazeFollow [53] and VideoAttentionTarget [15] to benchmark the AL methods: Entropy [57], MC-Dropout [26], Learning Loss [65], VAAL [58], AL-SSL [20] and our AL-GTD. Additionally, we refine the uncertainty definition introduced in UnReGa [8], which is used to improve source-free gaze estimation by minimizing uncertainty, and utilize these refinements as an AL acquisition function, establishing another novel AL methodology for gaze target detection.

Experiments show that AL-GTD can reduce the reliance on the size of labeled training data. For example, it can perform SOTA AUC results with only 50% of the GazeFollow [53] dataset's training split. Notably, it demonstrates promising results, i.e., nearing SOTA AUC, even when trained on only 20% of the full training set, indicating its proficiency in selecting highly informative training samples. Furthermore, our AL-GTD remarkably surpasses the SOTA gaze target detectors [59–61] when all are trained within a low-data regime (see Fig. 1 reflecting all these results). Comprehensive analyses confirm that our AL-GTD consistently outperforms other AL approaches and the ablation study proves the importance of each component of the proposed method.

The main contributions can be summarized as follows.
(1) This paper is the first attempt to achieve effective gaze target detection under the condition of limited training data.
(2) We introduce a novel deep AL approach for gaze target detection, outperforming its competitors and yielding robust results with fewer labeled training data. Our method achieves SOTA AUC performance by utilizing significantly less labeled training data, which is a performance level reached by [59–61] only when trained on the complete training dataset.
(3) We benchmark AL for gaze target detection by re-purposing several SOTA AL methods. We also present an additional AL acquisition function different from the proposed method by adjusting the approach in [8]. Such evaluations can be used as a reference by researchers interested in tackling the same task in the future.

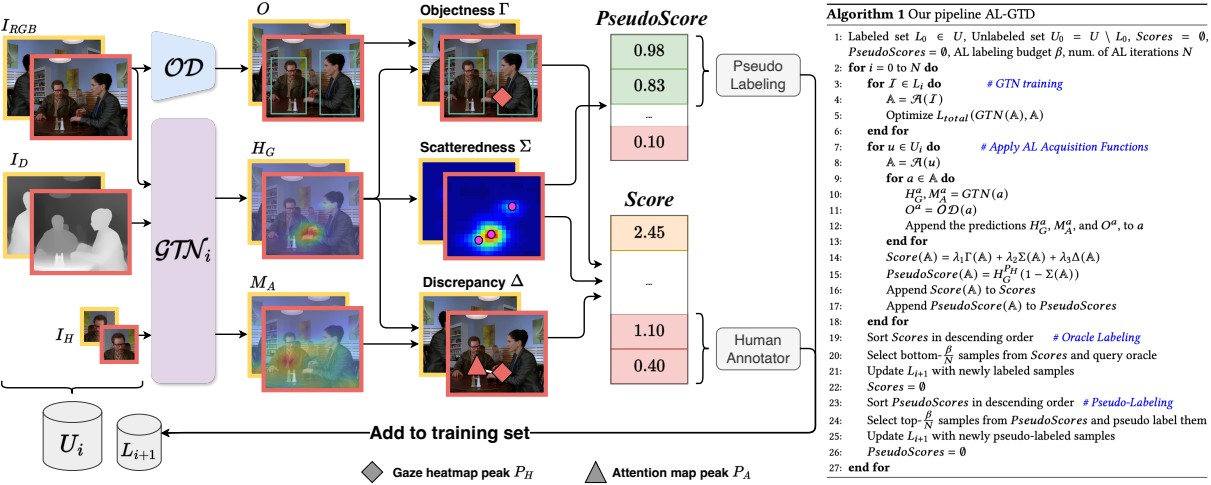

**Figure 2: The illustration and the pseudocode of our AL-GTD. We begin by obtaining the** augmented version $\mathbb{A}$ **from the** original version $\mathcal{I}$ **of unlabeled samples** $U_i$ **at the current AL cycle** $i$. $\mathcal{OD}$ **extracts relevant objects** $O$ **in the scene from** $I_{RGB}$, **while** $GTN$ **processes both** $I_{RGB}$ **and** $I_D$ **of the scene and the crop of the head of the person of interest** $I_H$. **From** $GTN$, **the attention map** $M_A$ **and gaze heatmap** $H_G$ **are obtained. The outputs of** $\mathcal{OD}$ **and** $GTN$ **are used to build the acquisition function (Eq. 6), composed of the objectness** $\Gamma$, **the scatteredness** $\Sigma$, **and the discrepancy** $\Delta$ **scores. The oracle annotates the most informative samples, while those with the lowest scatteredness (Eq. 7) are pseudo-labeled. Both the manually labeled samples by the oracle and the pseudo-labeled samples are added to the pool** $L_{i+1}$, **and** $GTN$ **is trained on the updated set. This process is repeated for a fixed number of iterations** $N$ **until the exhaustion of the labeling budget** $\beta$.

## 2 RELATED WORK

*Gaze Target Detection.* This task aims to determine the location that a person is looking at in a scene captured from a third-person perspective. The typical output of this task is a 2D heatmap, indicating the probability of where the person is looking in the scene while the gaze target is determined by the pixel coordinates corresponding to the highest value in the heatmap. To this end, [53] introduced the GazeFollow dataset and proposed a two-pathway approach. In that approach, one CNN-based pathway concentrates on extracting features from the head crop of the individual whose gaze is to be detected, while the other CNN-based pathway extracts features from the RGB scene image, with subsequent fusion of these features. The aforementioned two-pathway methodology is further advanced in [15], integrating spatiotemporal modeling for video-based gaze target prediction as well as introducing VideoAttentionTarget dataset. This concept is refined in subsequent works [4, 23, 34, 47, 59], some of which introduce a third pathway to incorporate the depth map of the scene image, which is estimated using a monocular depth estimator [23, 34, 47, 59]. Recently, Transformer-based methods are introduced [60, 61]. Such methods can simultaneously detect the gaze targets of multiple individuals in a scene. However, these models rely on larger training datasets, leading to performance degradation in low-data scenarios. To tackle this issue, we introduce AL-GTD, an innovative multimodal (RGB + depth) gaze target detection model specifically designed to facilitate Deep Active Learning in settings constrained by the amount of labeled training data.

*Deep Active Learning.* AL aims to optimize labeling costs by iteratively annotating (i.e., querying a human) [48] only those most informative samples from a large pool of unlabeled data. In recent years, special attention has been given to AL for a variety of tasks, with works addressing image classification [5, 10, 18, 25, 26, 28, 37–39, 48, 56, 58], object detection [13, 20, 35, 45, 55, 63], semantic segmentation [7, 27, 30, 37, 64], pedestrian detection in videos [2], person re-identification [43, 62], human hand pose estimation [29], video action detection [50], and temporal action localization [32].

These works can be categorized based on their acquisition function as: *i)* uncertainty-based, *ii)* diversity-based, or *iii)* committee-based, with also the possibility of hybrid methods [13, 20, 45, 47]. Uncertainty-based methods primarily depend on entropy [13, 20, 32, 45, 63, 64]. A clustering-based selection criterion was proposed in [50] to ensure diversity across samples while [65] introduced a Learning Loss module to predict the losses of unlabeled samples, which was later refined in subsequent works [10, 37]. On the other hand, Gal *et al.* [25, 26] presented deep Bayesian AL, which involves training a model with dropout layers and employing Monte Carlo dropout to approximate the sampling from the posterior. This method later became a mainstream baseline, as evidenced in [5, 13, 18, 20, 28, 38, 39, 45, 48, 56, 58]. Furthermore, Sinha *et al.* [58] integrated a Variational AutoEncoder (VAE) and a discriminator to establish an AL metric. Committee-based models, where ensembles are typically employed for sample set selection, also utilize a separate task classifier trained in a fully supervised manner [5, 29, 39, 48]. Recently, [20, 45] introduced a robustness score measuring an image's consistency and its augmented version. Using pseudo-labels is also very beneficial, as seen in [20]. However, the primary focus of such solutions lies in the classification head, overlooking regression problems such as gaze target detection. It is imperative to highlight that the domain of AL for gaze target detection remains unexplored.

## 3 METHOD

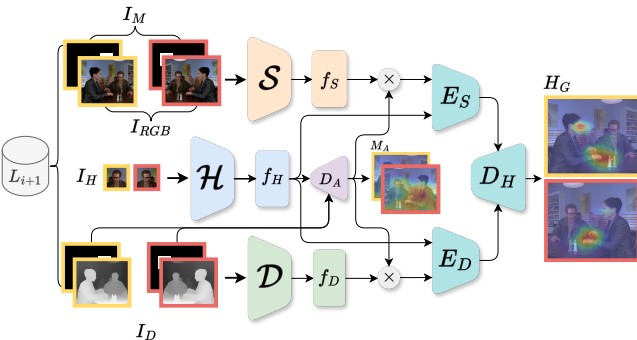

**Figure 3: Our proposed** $GTN$. $\mathcal{S}$ **and** $\mathcal{D}$ **process the scene RGB image** $I_{RGB}$ **and depth map** $I_D$, **respectively. The crop of the head of the person of interest** $I_H$ **is processed by a separate head branch** $\mathcal{H}$, **and** $D_A$ **projects the head features into the attention map** $M_A$. **The scene** $f_S$ **and depth** $f_D$ **features are multiplied by the attention map** $M_A$ **and processed by two separate encoders,** $\mathcal{E}_S$ **and** $\mathcal{E}_D$, **along with the head features** $f_H$. **Finally, the decoder** $D_H$ **processes the features of the encoders and generates the gaze heatmap** $H_G$. **To alleviate prediction inconsistency, we train on both the** original version $I$ **and the** augmented version $\mathbb{A}$ **of each labeled sample.**

Gaze target detection aims to predict the image coordinates where a person is looking in a scene captured from a third-person perspective. As outlined in previous works [23, 34, 47, 59], multi-modal information (*e.g.* the depth map) substantially enriches the gaze target detection performance. To accomplish this, we define a model $GTN$ that is fed with an input $I = \{I_{RGB}, I_D, I_H\}$, where $I_{RGB} \in \mathbb{R}^{W \times H \times 3}$ and $I_D \in \mathbb{R}^{W \times H}$ represents the RGB and depth image of the scene, and $I_H \in \mathbb{R}^{w \times h \times 3}$ denotes a $w \times h$ crop centered on the person's head whose gaze is subject to be predicted. The output $H \in \mathbb{R}^{W \times H}$ is the predicted gaze heatmap, where higher activation areas correspond to the area where the person is looking.

Our gaze target detector $GTN$ works within active learning such that, given an unlabeled dataset $U$ and a labeling budget $\beta$, the objective is to iteratively select the most informative samples from $U$ and annotate them using an oracle (*i.e.*, human) to create a new labeled set $L$ that maximizes the performance of $GTN$. Initially, a small random set of samples $L_0 \in U$ are labeled and are used to train a prior model $GTN_0$. Following this, $L$ is progressively populated over $N$ AL iterations. Thus, for each iteration $i \in [1, N]$, $GTN_{i-1}$ processes $U_{i-1} = U \setminus L_{i-1}$ and the predictions are evaluated by an acquisition function to determine the most informative samples, which are then labeled by an oracle. Meanwhile, we also pseudo-label the samples with the highest confidence, assuming that they are less informative samples or, in other words, the ones our $GTN$ can perform accurate gaze predictions. Both pseudo-labeled and human-annotated samples are added to the labeled set $L_i$. Finally, a new model $GTN_i$ is trained on $L_i$, and the process is repeated until the labeling budget $\beta$ is exhausted. This method, namely, AL-GTD, integrates supervised and self-supervised losses within a novel AL acquisition function to enable the usage of our gaze target detector under low-data regimes. An overview of AL-GTD is depicted in

Fig. 2, along with the associated pseudocode. Detailed explanations of its components, including $GTN$ (Fig. 3), are provided below.

### 3.1 Our Gaze Target Detector

To make the model more robust and impose a self-supervised consistency, which is used during AL, we also define a set of augmented inputs of $I$ as $\mathbb{A} = \mathcal{A}(I)$, with $\mathcal{A}$ being standard augmentations (*e.g.* random cropping, horizontal flipping, contrast/brightness changes, etc...). $I$ is processed by different branches which interact with each other to generate the output $H$. The scene branch $\mathcal{S}$ processes the image $I_{RGB}$ while the depth branch $\mathcal{D}$ processes the depth map $I_D$. Both inputs are enriched with the head mask $I_M$, which encodes the head's position in the image. The crop of the head of the person of interest $I_H$ is processed by a separate head branch $\mathcal{H}$, which extracts high dimensionality features $f_H$ about the head pose and gaze of the person. Furthermore, a separate module $D_A$ projects the concatenation of $f_H$ and $I_M$ into the attention map $M_A$, which encodes the probability of areas of the scene being gazed at by the person with only the head features and its position in the image. The RGB and depth features of the scene, denoted as $f_S$ and $f_D$, respectively, are combined with $M_A$ to prioritize areas where the gaze is likely to occur. These features are then processed by parallel encoders $E_S$ and $E_D$, projecting scene and head cues into two latent spaces. Finally, a decoder $D_H$ fuses the latent spaces above and produces the gaze heatmap $H_G$ focused on the person's gaze point.

### 3.2 Our Active Learning

Our AL pipeline comprises a novel acquisition function (Sec. 3.2.1), which evaluates the objectness determined by leveraging an object detector, the scatteredness of heatmap activations, and the discrepancy between $M_A$ and $H_G$. Additionally, we incorporate pseudo-labeling (Sec. 3.2.2) and self-supervised learning (Sec. 3.2.3) into our approach.

*3.2.1 Acquisition function.* Recall that the objective of an acquisition function is to identify which samples are the most informative and can enhance the network's predictions. AL literature [13, 20, 32, 45, 63, 64] shows that high entropy samples (representing low-confidence) are expected to provide more information than low ones. However, in the context of gaze target detection, highly confident low entropy predictions can result in incorrect gaze heatmaps, particularly in complex scenes where multiple people and objects exist. Conversely, images with low confident predictions and high entropy may indicate poor model robustness and, if labeled, they might not bring new information to the existing network. Therefore, an acquisition function for gaze target detection must address several perspectives. We propose three properties that determine the informativeness of images to be labeled, described as follows.

*Objectness.* Inspired by studies demonstrating that individuals often gaze at living or non-living *objects* during social and physical interactions [12, 40, 46], we incorporate the use of an object detector $O\mathcal{D}$ under the assumption that it could enhance the informativeness of samples for gaze target detection within AL. Due to how the scene branch $\mathcal{S}$ is pre-trained, elements in the scene can influence $D_H$ to predict the gaze heatmap onto objects. This behavior

potentially leads to poor generalization, especially when many objects are visible but the person is looking elsewhere. We empirically find that objects in the foreground shift the activation of the gaze heatmap toward the object's center, ignoring cues provided by the attention map. This detector takes the scene image as input and predicts a set of objects $O = O\mathcal{D}(I_{RGB})$. Each object prediction $o \in O$ consists of a bounding box $b = \{c_x, c_y, w, h, k\}$, with $c_x, c_y, w, h$ denoting the center coordinates, width, and height of the object, respectively, while $k \in \mathbb{R}^K$ represents the probability distribution across the $K$ classes considered. To discourage the network from taking shortcuts into foreground objects, we detect objects being targeted by the gaze heatmap and calculate the max confidence

$$\gamma(O, H_G) = \max(\{c_o \, \mathbb{1}_{O_H}(o) : o \in O\}), \tag{1}$$

where $O_H \subseteq O$ is the set of objects such that the peak activation $P_H$ of the heatmap $H_G$ lies inside their bounding boxes, $c_o$ is the confidence of the object, and $\mathbb{1}$ is the indicator function of $O_H$. The objectness score is calculated as the maximum confidence across multiple augmentations of the same scene, *i.e.*:

$$\Gamma(\mathbb{A}) = \max(\{\gamma(O^a, H_G^a) : a \in \mathbb{A}\}), \tag{2}$$

where $(\cdot)^a$ represents the prediction for sample $a$.

*Heatmap activation scatteredness estimation.* The peak activation points of the gaze heatmap $H_G$ should ideally be densely positioned on a small region, with values progressively increasing as we reach the peak of the heatmap. Conversely, sparse peak activation points indicate network uncertainty. Let $\rho: \mathbb{R}^{n \times n} \to \mathbb{R}^{n^2 \times 2}$ be the function that takes a positive-defined matrix and ranks in descending order its elements and returns the coordinates of the cells. We discretize the heatmap into $B$ bins, obtain the heatmap peaks coordinates in descending order as $\pi = \rho(\cdot)$, and compute the scatteredness of the activations as:

$$\sigma(\pi) = \frac{1}{P} \sum_{p=1}^{P} ||\pi_1 - \pi_{p+1}||_2, \tag{3}$$

where $\pi_1$ is the maximum activation point of the heatmap and $\pi_{p+1}$ are the coordinates of the $p$-most highest and farthest activation point from $\pi_1$. The scatteredness function iteratively finds the highest $P$ activation points farthest from the peak and calculates their distance from it. The final scatteredness score is calculated as:

$$\Sigma(\mathbb{A}) = \max(\{\sigma(\rho(H_G^a)) : a \in \mathbb{A}\}), \tag{4}$$

with high values indicating network uncertainty in at least one of the augmentations $a$.

*Discrepancy between attention map and gaze heatmap.* The difference between the attention map $M_A$ and gaze heatmaps $H_G$ is a useful measure of model uncertainty. In effective gaze target detection networks, $M_A$ and $H_G$ must agree in the *direction (orientation)* and *location* of gaze, with the latter being a more refined version of the former. To estimate the differences between them, we compute the distance among the peak activation points $\delta(M_A, H_G) = ||P_A - P_H||_2$, where $P_A$ and $P_H$ are the positions in the 2D image space of the highest activation point of the attention map and gaze heatmap, respectively.

To address model robustness, we estimate the disagreement across multiple augmentations of the same scene and calculate the maximum distance among augmentations as:

$$\Delta(\mathbb{A}) = \max(\{\delta(M_A^a, H_G^a) : a \in \mathbb{A}\}). \tag{5}$$

Informative samples have a high value for $\Delta(\mathbb{A})$, and labeling them allows the network to reduce the discrepancy between the saliency map and gaze heatmap.

The three above criteria are aggregated to identify those samples whose attention maps differ from the predicted gaze heatmap, that contain objects with high confidence included in the heatmap, and whose activation points are sparse. The final *Score* of a sample is calculated as:

$$Score(\mathbb{A}) = \lambda_1 \Gamma(\mathbb{A}) + \lambda_2 \Sigma(\mathbb{A}) + \lambda_3 \Delta(\mathbb{A}), \tag{6}$$

where $\lambda_1$, $\lambda_2$, and $\lambda_3$ are the learnable weights.

After assigning scores to all samples in the unlabeled set, we label the top $\beta/N$ samples with the highest scores. The process is repeated for $N$ active learning iterations.

*3.2.2 Gaze heatmap pseudo-labeling.* We contend that it is essential for the gaze network to come across representative and easily detectable samples in order to avoid distribution shifts. At the same time, our goal is to avoid labeling confident samples to save labeling resources. As a solution, we suggest utilizing a pseudo-labeling technique, where the network trained in the previous AL cycle generates pseudo-labels for the network currently undergoing training. We calculate a labeling score for each unlabeled sample as

$$PseudoScore(\mathbb{A}) = H_G^{P_H}(1 - \Sigma(\mathbb{A})), \tag{7}$$

where $H_G^{P_H}$ is the peak value of the gaze heatmap, and we pseudo-label samples with the highest scores up to a fixed percentage per iteration.

*3.2.3 Supervised and self-supervised training.* The network is end-to-end trained on both labeled and pseudo-labeled samples by minimizing the Mean Squared Error loss between the predicted heatmap $H_G$ and $\tilde{H}_G \sim \mathcal{N}(\tilde{P}_H, \Sigma)$, with $\tilde{P}_H$ being the ground-truth/pseudo-labeled gaze point and $\Sigma$ being a positive covariance matrix. In addition to this supervised loss, we incorporate also a self-supervised loss to ensure consistency across multiple augmentations $\mathbb{A}$ of the input. The self-supervised learning (SSL) between augmentations of the same image allows us to increase the robustness of the network and reduce the probability of assigning a high $Score(\mathbb{A})$ to easy, non-informative samples, whose discrepancy and scatteredness scores are high because of poor network generalization. We define the consistency loss $L_c$ among predictions as $L_c(a, a') = ||H_G^a - \mathcal{A}^{-1}(H_G^{a'})||_2$ where $\mathcal{A}^{-1}$ is the inverse augmentation function that aligns the predictions and allows us to effectively calculate the localization error among augmentations. After predicting the gaze heatmap $H_G$ for all $a \in \mathbb{A}$, we compute the supervised loss with the ground-truth gaze heatmap $\tilde{H}_G$ as $L_h(a) = ||H_G^a - \tilde{H}_G^a||_2$. We then compute the total loss as:

$$L_{total}(\mathbb{A}) = \sum_{a \in \mathbb{A}} \sum_{a' \in \mathbb{A} \backslash a} L_c(a, a') + \sum_{a \in \mathbb{A}} L_h(a). \tag{8}$$

Francesco Tonini, Nicola Dall'Asen, Lorenzo Vaquero, Cigdem Beyan, and Elisa Ricci

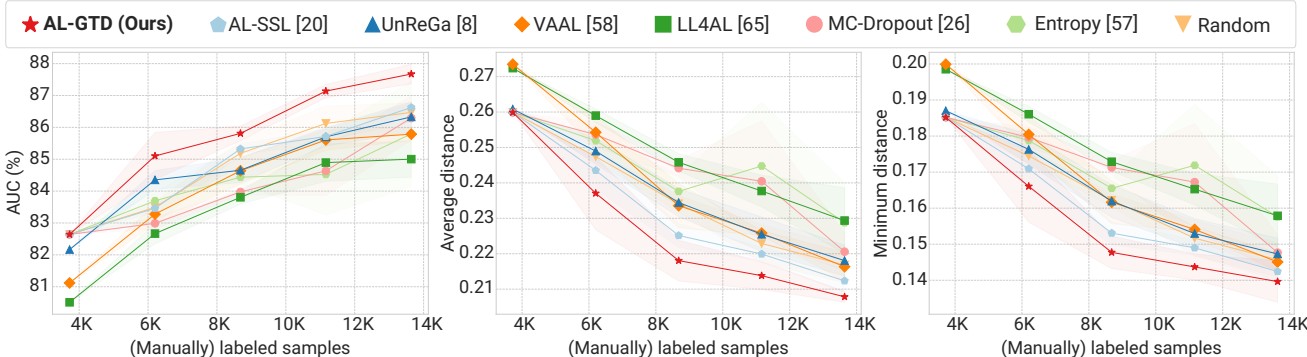

**Figure 4: Comparisons among AL methods on the GazeFollow [53] dataset. Left: Area Under the Curve (AUC) of the predicted gaze heatmap w.r.t. the ground truth (GT). Center and right: average and minimum distance between GT and the predicted gaze point. Our method, AL-GTD, consistently surpasses random sampling and other AL methods, demonstrating superior performance even with a small initial training dataset (3.7K samples, ~3% of the original train split).**

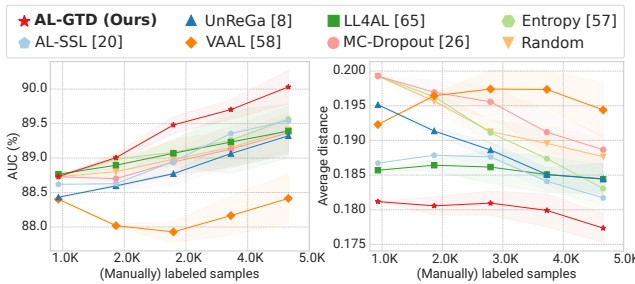

**Figure 5: Comparisons among AL methods on the VideoAttentionTarget [15] dataset. Left: Area Under the Curve (AUC) of the predicted gaze heatmap w.r.t. the GT. Right: average distance between GT and predicted gaze point.**

## 4 EXPERIMENTS

We evaluate all methods using the standard gaze target detection benchmarks GazeFollow [53] and VideoAttentionTarget [15]. **Gaze-Follow** is a large-scale image dataset comprising over 122K images, featuring annotations for gaze and head locations for more than 130K individuals. On the other hand, **VideoAttentionTarget** consists of YouTube video clips, each up to 80 seconds long, with 110K frame gaze annotations and their corresponding head locations. For this dataset, we follow the common practice of sampling one image every 20 frames [4, 15, 23, 34, 59–61]. To measure the performance of the algorithms, we use the standard **evaluation metrics** for gaze target detection [14, 15]. The **Area Under the Curve (AUC)** evaluates the confidence of the predicted gaze heatmap w.r.t. the ground truth (GT), while **Distance (Dist.)** represents the $\mathcal{L}_2$ distance between the GT gaze point and the location on the predicted gaze heatmap with maximum confidence. For GazeFollow, it is customary to report both the minimum and average distances, while for VideoAttentionTarget, only the average distance is reported due to the presence of a single GT point during evaluation.

### 4.1 Implementation Details

The scene $\mathcal{S}$ and depth $\mathcal{D}$ branches consist of a ResNet50 pre-trained on RGB and depth scene classification [66], respectively.

The object detector $\mathcal{OD}$ is based on DETR [67] and pretrained on the COCO [42] dataset. Following earlier works [4, 23, 34, 59, 60], we employ a monocular depth estimation network [51] to extract depth maps from the input images. The head branch $\mathcal{H}$ is built upon a ResNet50 backbone pretrained on the Eyediap dataset [24] and includes a feature projector $D_A$ to produce the attention map $H_c$ from head features $f_H$. We train $\mathcal{S}$, $\mathcal{D}$, and $\mathcal{H}$, as well as $D_A$, $E_S$, $E_D$, and $D_H$ with an Adam optimizer and a $2.5 \times 10^{-4}$ learning rate while keeping $\mathcal{OD}$ frozen. On GazeFollow [53], we bootstrap our prior model $GTN_0$ with 3.7K randomly sampled images from the training set (~3% of the original training split). Subsequently, we conduct five AL iterations using AL-GTD, where we label 2.5K images (~2% of the original training split) and pseudo-label 2.5K additional images per iteration. On VideoAttentionTarget [15], we bootstrap $GTN_0$ with 932 randomly selected images and label 932 frames per iteration. We perform four AL iterations and use the pseudo-labeling setup of the GazeFollow experiments.

***Other Methods.*** We compare AL-GTD with random sampling as well as AL methods such as Entropy [57], LL4AL [65], VAAL [58], AL-SSL [20], MC-Dropout [26], and the committee-based UnReGa [8]. The implementation details of the baselines are given in Supp. Mat. All baselines are trained and evaluated using the same configurations employed for AL-GTD, including the optimizers, learning rate, AL budget, and the initial subset randomly selected for training. At the start of each AL iteration, we reload the initial pre-trained weights to encourage the model to generalize on the updated data distribution, following common implementation, *e.g.*, [13, 20, 45].

### 4.2 Results

The main objective of this work is to decrease reliance on extensive amounts of labeled training data for gaze target detection. Our method, AL-GTD, accomplishes SOTA results of 92.2% AUC for GazeFollow dataset [53] using only 50% of the total training samples, rather than utilizing the entire training dataset, as in [59–61]. This is also illustrated in Fig. 1 in detail. Furthermore, this percentage decreases to 40% of the training set for VideoAttentionTarget dataset [15], in which AL-GTD achieves 93.5% AUC, once again

**Table 1: Ablation study on GazeFollow [53] in terms of AUC (best in bold) for the effect of SSL, and the components of the AL acquisition function: objectness ($\Gamma$), discrepancy ($\Delta$), and scatteredness ($\Sigma$). We also report the results of random sampling and AL-SSL [20] for the reader's reference. Note that the cycle with 3.7K samples represents the initial training, where no AL is applied. Therefore, the results are equal for all ($AUC = 82.64 \pm 0.00$).**

| SSL | $\Gamma$ | $\Delta$ | $\Sigma$ | 6.2K | 8.7K | 11.2K | 13.7K |
|---|---|---|---|---|---|---|---|
| ✗ | ✓ | ✓ | ✓ | 83.18 ± 0.22 | 84.85 ± 0.43 | 84.98 ± 0.51 | 85.60 ± 0.35 |
| ✓ | ✗ | ✓ | ✓ | 83.65 ± 0.14 | 85.32 ± 0.58 | 86.52 ± 0.31 | 87.20 ± 0.07 |
| ✓ | ✓ | ✗ | ✓ | 83.85 ± 0.46 | 85.74 ± 0.24 | 86.68 ± 0.28 | 86.91 ± 0.16 |
| ✓ | ✓ | ✓ | ✗ | 84.39 ± 0.72 | 85.47 ± 0.41 | 86.83 ± 0.64 | 87.37 ± 0.49 |
| ✓ | ✓ | ✗ | ✗ | 84.10 ± 0.62 | 85.42 ± 0.30 | 86.81 ± 0.11 | 86.82 ± 0.85 |
| ✓ | ✓ | ✓ | ✓ | **85.10 ± 0.74** | **85.81 ± 0.23** | **87.14 ± 0.21** | **87.67 ± 0.31** |
| Random | | | | 83.49 ± 0.17 | 85.18 ± 0.39 | 85.95 ± 0.25 | 86.46 ± 0.29 |
| AL-SSL [20] | | | | 83.45 ± 0.49 | 85.32 ± 0.15 | 85.71 ± 0.23 | 86.62 ± 0.11 |

proving the effectiveness of our AL acquisition function. Importantly, AL-GTD can reach satisfactory results, i.e., 90.3% AUC in GazeFollow [53] by being trained only 20% of the training data while this percentage is 10% of the training data for VideoAttention-Target [15] in which AL-GTD yields 92.5% AUC. We compare the performance of our AL-GTD against other AL methods in Sec. 4.2.1. We present an ablation study demonstrating the importance of each component of our AL acquisition function (Sec. 4.2.2) and examine the effect of pseudo-labeling (Sec. 4.2.3). Additionally, we provide comparisons between AL-GTD and other AL methods when semi-supervised learning (SSL) is adapted to them (Sec. 4.2.4). Following that, we compare AL-GTD with SOTA gaze target detectors under the condition of limited training data in Sec. 4.2.5. Finally, we display some qualitative results in Sec. 4.2.6.

*4.2.1 Comparisons with other AL methods.* Figs. 4 and 5 compare AL-GTD against its counterparts and random sampling for GazeFollow [53] and VideoAttentionTarget [15], respectively. These results together with the number of parameters and the elapsed times for training and inference, are available in tabular format in the Supp. Mat. The results demonstrate that AL-GTD consistently outperforms the random sampling and all other methods even from the initial cycle of AL in which the used training data is really small. Overall, the second-best performing method is AL-SSL [20]. While it is challenging to pinpoint a clear winner between AL-SSL [20] and other methods, AL-SSL [20] tends to outperform them as the size of the training data increases. However, it never manages to surpass our method, as evidenced in Fig. 1. On average, the worst performing methods are VAAL [58] and LL4AL [65].

*4.2.2 Ablation Study.* The contributions of the components of our AL acquisition function (i.e., discrepancy $\Delta$, scatteredness $\Sigma$, and objectness $\Gamma$) and SSL are reported in Tab. 1 in terms of AUC for the GazeFollow dataset. Refer to the Supp. Mat. for the Dist. results, which back up the conclusions drawn from the AUC-based analysis.

Our first objective is to understand the effect of SSL by removing it from our pipeline (*Row 1 vs. Row 6*), which results in a decrease in performance (sometimes even slightly more than 2% AUC) in every AL cycle. We then assess the impact of each acquisition component by removing them one at a time while keeping SSL (*Rows 2-4*). In these instances, the results reveal that there is no clearly superior

**Table 2: Performance of AL-GTD associated with different numbers of samples pseudo-labeled. The results correspond to 13.7K manually annotated samples from the GazeFollow [53] dataset.**

| Percentage | AUC ↑ | Avg. Dist. ↓ | Min. Dist. ↓ |
|---|---|---|---|
| 0% | 87.19 ± 0.11 | 0.210 ± 0.005 | 0.141 ± 0.005 |
| 0.1% | 87.49 ± 0.34 | 0.213 ± 0.003 | 0.144 ± 0.003 |
| 2% | **87.67 ± 0.31** | **0.208 ± 0.002** | **0.140 ± 0.006** |
| 5% | 86.57 ± 0.34 | 0.223 ± 0.008 | 0.153 ± 0.007 |

**Table 3: AUC scores of AL-GTD on GazeFollow [53] based on pseudo-labeling criteria used. Note that the cycle with 3.7K samples represents the initial training stage, where no AL is applied. Therefore, the results are equal for all.**

| Criteria | 3.7K | 6.2K | 8.7K | 11.2K | 13.7K |
|---|---|---|---|---|---|
| $Score(\mathbb{A})$ | 82.64 ± 0.00 | 84.68 ± 0.03 | 85.65 ± 0.07 | 86.68 ± 0.45 | 87.28 ± 0.36 |
| $\Gamma(\mathbb{A})$ | 82.64 ± 0.00 | 83.51 ± 0.04 | **86.53 ± 0.01** | 86.82 ± 0.56 | 87.53 ± 0.35 |
| $\Delta(\mathbb{A})$ | 82.64 ± 0.00 | 84.14 ± 0.43 | 85.11 ± 0.34 | 86.70 ± 0.23 | 87.27 ± 0.37 |
| $\Sigma(\mathbb{A})$ | **82.64 ± 0.00** | **85.10 ± 0.74** | 85.81 ± 0.23 | **87.14 ± 0.21** | **87.67 ± 0.31** |

criterion among the three. However, on average, the removal of the objectness ($\Gamma$) criterion results in a slightly greater decrease in performance compared to the removal of the others one at a time. Consequently, we also tested retaining the SSL and $\Gamma$ criteria while removing all other criteria (*Row 5*). Also, in this case, performance drops in every AL cycle. Therefore, we can conclude that each criterion contributes importantly and using them together yields the best gaze target detection results. Furthermore, it is noteworthy that all combinations still outperform the runner-up method [20] and random sampling.

*4.2.3 The effect of pseudo-labeling.* Tab. 2 presents the performance of AL-GTD across various numbers of samples pseudo-labeled. These results correspond to 13.7K manually annotated samples from the GazeFollow dataset [15], while the percentages for pseudo-labeling are relative to the full size of the training data. One can observe that pseudo-labeling overall contributes to performance enhancement. For instance, with 0.1% or 2% pseudo-labeling, improvements are noticeable compared to not using pseudo-labeling. Conversely, increasing the percentage of pseudo-labeling, such as up to 5%, may lead to performance decreases in the low data regime training. Based on these findings, we choose to conduct our main experiments with 2%, which is equal to the amount of data we request human annotators to manually label.

Furthermore, as explained in Sec. 3.2.2 and Eq. 7, the samples to be pseudo-labeled are chosen based solely on the $\Sigma$ criterion, which has been empirically found to be the most effective in low-data regimes. The corresponding results are provided in Tab. 3, indicating that when we use a $Score(\mathbb{A})$ equivalent to the criterion for selecting samples to be manually labeled or replace $\Sigma$ in Eq. 7 with $\Gamma$ or $\Delta$, the AUC values decrease.

*4.2.4 The effect of SSL.* We injected our SSL to Entropy [57], UnReGa [8] and MC-Dropout [26] and compare them with SSL-based methods: AL-GTD and AL-SSL [20] in Tab. 4 in terms of AUC. One can observe that it is possible for SSL-based Entropy, MC-Dropout and UnReGa to surpass AL-SSL [20] while AL-GTD still performs the best out of all. The results from other metrics and AL cycles consistently confirm the performance of AL-GTD (see Supp. Mat.).

**Table 4: Comparisons among SSL-based methods for different cycles of AL. Note that the cycle with 3.7K samples represents the initial training stage, where no AL is applied. Thus, the results are equal for methods whose loss function is the same.**

| Method + SSL | 3.7K | 6.2K | 8.7K | 11.2K | 13.7K |
|---|---|---|---|---|---|
| Entropy [57] | **82.64 ± 0.00** | 84.09 ± 0.47 | 85.41 ± 0.15 | 86.23 ± 0.49 | 86.48 ± 0.96 |
| MC-Dropout [26] | **82.64 ± 0.00** | 83.77 ± 0.32 | 85.66 ± 0.33 | 86.46 ± 0.26 | 86.79 ± 0.27 |
| UnReGa [8] | 81.79 ± 0.00 | 83.73 ± 0.44 | 85.57 ± 0.09 | 86.25 ± 0.13 | 87.06 ± 0.02 |
| AL-SSL [20] | **82.64 ± 0.00** | 83.46 ± 0.49 | 85.32 ± 0.15 | 85.71 ± 0.24 | 86.62 ± 0.12 |
| **AL-GTD (Ours)** | **82.64 ± 0.00** | **85.10 ± 0.74** | **85.81 ± 0.23** | **87.14 ± 0.21** | **87.67 ± 0.31** |

**Table 5: Comparisons between our AL-GTD and SOTA gaze target detectors trained under low data regimes (i.e. 13.7K and 62K samples, which correspond to ~10% and ~50% of the dataset, respectively) on GazeFollow [53] using our AL-GTD's sample selection. ⋆ denotes random sample selection.**

| | AUC ↑ | | Avg. Dist. ↓ | | Min. Dist. ↓ | |
|---|---|---|---|---|---|---|
| | 10% | 50% | 10% | 50% | 10% | 50% |
| [59] (ICMI 2022)⋆ | 85.88 | 90.51 | 0.225 | 0.157 | 0.154 | 0.099 |
| [59] (ICMI 2022) | 86.72 | 91.04 | 0.222 | 0.153 | 0.150 | 0.090 |
| [61] (CVPR 2022) | 67.30 | 80.30 | 0.299 | 0.213 | 0.226 | 0.151 |
| [60] (ICCV 2023) | 77.00 | 84.10 | 0.250 | 0.160 | 0.183 | 0.105 |
| **AL-GTD (Ours)** | **87.67** | **92.21** | **0.208** | **0.147** | **0.140** | **0.084** |

*4.2.5 Comparisons with SOTA Gaze Target Detectors.* We compare the performance of AL-GTD and SOTA gaze target detectors [59–61] under the limited training data regime in Tab. 5. In this context, we present results obtained from two sets of samples: one comprising 13.7K samples (~10% of the entire training data), which we deem manageable for annotation in real-world scenarios; and another comprising 62K samples (~50% of the whole training set), corresponding to the set on which AL-GTD achieves SOTA AUC performance. These 13.7K and 62K samples are drawn from the GazeFollow dataset [53] selected by AL-GTD's acquisition function.

The corresponding results demonstrate that Transformer-based models [60, 61] consistently exhibit remarkably lower performance across all metrics and subset sizes compared to AL-GTD. Such differences are attributed to the high volume of training data required by Transformers. Instead, CNN-based SOTA [59] demonstrates better performances compared to [60, 61], although it still falls short of AL-GTD's performances. We assert that while all other methods may encounter challenges with limited training data, AL-GTD's SOTA AUC performance with 50% of the training data represents an important success. Such achievement is attainable by others only with access to the full training dataset. In terms of Dist. metrics AL-GTD trained with 50% of the training data surpasses [59]'s performance when trained on the full training set. However, all methods require more data to achieve significantly low Dist. scores.

We also investigate the quality of the data selected by AL-GTD by comparing the performance of [59] on an equal amount of randomly selected data (shown with ⋆ in Tab. 5). Such results demonstrate that the data selected by AL-GTD is more informative for [59], showing that AL-GTD can be used for training data curation.

*4.2.6 Qualitative Results.* We visualize gaze heatmaps generated by AL-GTD, the best-performing AL counterpart [20], and random selection in Fig. 6. For additional qualitatives, refer Supp. Mat.

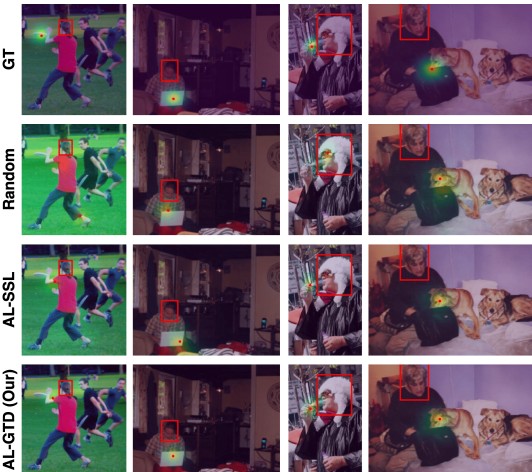

**Figure 6: Gaze heatmaps produced by AL-GTD and others.**

## 5 DISCUSSIONS AND CONCLUSIONS

We have presented AL-GTD, a novel multimodal gaze target detection model that incorporates active learning (AL). Our proposal achieves SOTA AUC performance with just 40-50% of the usual training data and is competitive w.r.t. the current best-performing models even when trained on a mere 10-20% of the full training dataset. This highlights the ability of AL-GTD to effectively choose highly informative samples during training. AL-GTD integrates supervised and self-supervised losses (SSL) within a novel AL acquisition function while also employing pseudo-labeling to effectively mitigate potential distribution shifts during training. It demonstrates superior performance over all SOTA AL methods, even when those AL methods are combined with the same SSL strategy and pseudo-labeling employed by ours. This shows the effectiveness of our primary technical novelty, the AL acquisition function, in enabling rapid learning.

Despite setting a new SOTA in gaze target detection within a reduced labeled data setting, we still believe that the Distance metric can be improved. One way to tackle this is by integrating a loss function specifically optimized for calculating the distance between the ground truth and predicted gaze points. An effective strategy could entail estimating the peak from the predicted heatmap in a differentiable manner [44] and computing the $L1$ distance w.r.t. the ground truth. Furthermore, Transformer-based gaze target detectors have shown limited effectiveness in low-data scenarios, as demonstrated in our study. Surprisingly, there are currently no Deep AL methods that utilize Transformer-based backbones. A potential avenue for further research could involve replacing the *GTN* in AL-GTD with a Transformer-based gaze target detector, incorporating additional modules to enhance the Transformer's performance in low-data environments. Lastly, while our gaze target detection model incorporates both scene features and depth map features equally to establish the scatteredness and discrepancy criteria, we aim to further capitalize on the consistencies between these modalities. We will investigate the definition of explicit rules that govern the relationships across these different modalities, aiming to incorporate these rules into our AL acquisition function for improved performance.

# ACKNOWLEDGMENTS

We thank CINECA and the ISCRA initiative for the availability of high-performance computing resources. This work was supported by the EU H20202 SPRING (No. 871245) project, the EU Horizon ELIAS (No. 101120237) project, the MUR PNRR project FAIR - Future AI Research (PE00000013) funded by the NextGenerationEU, and the PRIN LEGO-AI (Prot. 2020TA3K9N) project. This work was carried out in the Vision and Learning joint laboratory of FBK and UNITN.

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
