# OpenReview forum: "AL-GTD: Deep Active Learning for Gaze Target Detection"
_acmmm.org/ACMMM/2024/Conference — MM2024 Poster_

### Official Review · Reviewer_7USL · 2024-05-12

**Rating:** 5
**Confidence:** 3

**Summary:**

This paper focuses on the task of gaze target detection which aims to locate the target region where a person is looking in image or video scenarios. To tackle the issue of current methods relying on extensive labeled data, this paper proposes an active learning method that integrates supervised and self-supervised losses within a novel sample acquisition function. In addition, this paper also introduces pseudo-labeling to tackle the distribution shifts in the training phase. Experiments on the benchmark datasets validate the effectiveness of the proposed method when trained within a low-data regime.

**Strengths:**

- The utilization of active learning to realize gaze target detection in limited training data is novel.
- Extensive experiments are conducted on the benchmark dataset to validate the effectiveness of active learning for gaze target detection.
- This paper is well organized and easy to read.

**Limitations:**

- The paper utilizes pseudo-labeling to augment the training dataset but lacks a detailed explanation of the protocol for generating these pseudo-labels and how to avoid distribution shifts.
- The standard augmentations (e.g., random cropping, horizontal flipping) are adopted to augment the inputs. However, these augmentation techniques are typically used for image recognition tasks, and their applicability and effectiveness specifically for gaze target detection may not be optimal. Could the authors discuss whether any specialized augmentation strategies are tailored to the unique challenges of gaze target detection?

**Suitability:**

2

---

### Official Review · Reviewer_Gugx · 2024-05-20

**Rating:** 3
**Confidence:** 3

**Summary:**

The paper introduces AL-GTD, a novel approach for gaze target detection that reduces reliance on large labeled datasets by integrating supervised and self-supervised losses within an innovative sample acquisition function for active learning. AL-GTD achieves superior performance compared to state-of-the-art gaze target detectors, using only 40-50% of the training data, and quickly reaches satisfactory results with just 10-20% of the data, highlighting the effectiveness of its acquisition function in selecting the most informative samples.

**Strengths:**

The proposed methodology addresses the challenge of relying on large labeled datasets by combining supervised and self-supervised losses within an innovative sample acquisition function for active learning. This approach is significant because obtaining large amounts of training data is often impractical. While this method is relevant to various vision problems, it would be nice to see (a discussion or explanation on broader impact) how this concepts would translate to other CV challenges.

**Limitations:**

1. Although the author highlighted their contributions in three major points, the core contribution is the use of limited datasets to achieve or surpass the state-of-the-art (SOTA).

2. There are inconsistencies in the paper:

    - What is **V** in line 502 and Equation (2)?
    - The use of vague terms like **reaches satisfactory performance** (Abstract Line 20-21) and other parts of the paper. What is satisfactory performance? How is this defined?
    - In Sec 3.2.1, Lines 425-428, the author should cite relevant evidence or provide an explanation for why **highly confident low entropy predictions can result in incorrect gaze heatmaps**.

3. It is unclear how scatterdness is computed. How do you define heatmap peaks given there could be multiple regions of interest?

4. In Figure 4, the experimental results (AUC curve) show that the error margin for AL-GTD is reasonably higher for a 6K sample size but becomes considerably consistent or slimmer for UnReGa. This behavior could imply inconsistency in the proposed method. Is there any study regarding such variability? Is it due to the fact that some random samples can have complex scenes?

5. In Table 1, the ablation study shows that for the limited sample size (a strong aspect of the proposed work), objectness, discrepancy, and scatterdness are indifferent, suggesting that 3.7K samples are too few for the proposed method, and notable differences appear at 6.2K. Is there any study indicating the optimal limited size for this proposed method?

6. From the complex scenes provided in the supplementary material, the method seems best suited for simpler scenes. However, there might be chances for improvement by training with larger datasets to achieve better results. Is there any ablation study on qualitative results for complex scenes comparing models trained with an optimally limited data size versus 50% or 100% of the data? Are we sacrificing the model's performance on complex scenes?

**Suitability:**

3

---

### Official Review · Reviewer_7HFr · 2024-05-24

**Rating:** 5
**Confidence:** 3

**Summary:**

This paper proposes a novel active learning strategy to reduce label dependency on training data for gaze estimation. The proposed framework utilizes pseudo labelling technique to mitigate the distribution shift in the data.

**Strengths:**

The active learning framework seems promising as it has the potential to address the label based noise usually present in the gaze data.

The experiments have been conducted on GazeFollow and VideoAttentioTarget datasets. The proposed algorithm reaches SOTA with only 50% of the training data.

The quantitative analysis and ablation studies are thorough. It shows that the ssl strategy, discrepancy, scatterness and objectiveness of the proposed pipeline plays crucial role in overall performance.

**Limitations:**

Just a minor point, figure 6 is difficult to understand. Please put a gaze vector or any other way to show the concerned subject and subject specific target.

**Suitability:**

3

---

### Official Review · Reviewer_LqkH · 2024-05-25

**Rating:** 4
**Confidence:** 2

**Summary:**

The paper presents a novel approach to gaze target detection that aims to reduce reliance on extensive labeled datasets. The proposed method, AL-GTD, integrates supervised and self-supervised learning with a novel sample acquisition function for active learning. The approach also employs pseudo-labeling to handle distribution shifts during training. AL-GTD achieves state-of-the-art performance using only 40-50% of the training data required by existing methods. The paper provides a comprehensive experimental analysis, demonstrating that AL-GTD outperforms other active learning methods and traditional gaze target detectors in low-data regimes.

**Strengths:**

1. This work is a meaning attempt to achieve effective gaze target detection under the condition of limited training data.
2. AL-GTD achieves SOTA performance using significantly less labeled data compared to existing methods, demonstrating efficiency in data usage.
3. The paper is well-written and easy to follow. The diagrams illustrate motivations and the details well.

**Limitations:**

1. Please provide a detailed explanation of the structure of GTN in the Method section, such as the specific network architecture of the three backbones S, H, and D.
2. It is recommended to include more gaze heatmap visualizations and qualitative analyses to validate the effectiveness of the proposed method and compare it with other SOTA methods. These analyses could be included in the main text or supplementary materials.
3. Please cite some of the latest gaze / visual attention studies from 2024.
[AAAI2024] Gaze Target Detection by Merging Human Attention and Activity Cues
[CVPR2024] Learning from Observer Gaze: Zero-Shot Attention Prediction Oriented by Human-Object Interaction Recognition
4. Some formulas lack punctuation, such as formulas (2), (3), (4), (5), and (6).
If the authors address my concerns, I would be happy to raise my score.

**Suitability:**

2

---

### Meta-Review · Area_Chair_3aGD · 2024-06-30

**Recommendation:** Accept (Poster)
**Confidence:** 5

**Metareview:**

All the reviewers are satisfied with the response. The final paper should incorporate the author's valid responses. I am delighted to recommend the acceptance of this paper.

---

### Meta-Review · Senior_Area_Chairs · 2024-07-10

**Recommendation:** Accept (Poster)
**Confidence:** 5

**Metareview:**

This paper received mixed ratings initially. After rebuttal, all the reviewers tend to accept the paper. SAC and AC agree with reviewers and recommend accepance.